# Parents' experiences with a sick or injured child during the COVID-19 lockdown: an online survey in the Netherlands

Chantal D Tan ![ORCID],[1] Eveline K Lutgert,[1] Sarah Neill,[2] Rachel Carter,[2] Ray B Jones ![ORCID],[2] Jade Chynoweth,[2] Dorine M Borensztajn ![ORCID],[1] Monica Lakhanpaul ![ORCID],[3] Henriette A Moll ![ORCID] [1]

¹General Paediatrics, Erasmus MC - Sophia Children's Hospital, Rotterdam, The Netherlands
²Faculty of Health, University of Plymouth, Plymouth, UK
³Integrated Community Child Health Population, Policy & Practice Department, GOS Institute of Child Health, University College London, London, UK

**Correspondence to**
Chantal D Tan;
c.tan@erasmusmc.nl

## ABSTRACT

**Objective** To assess the impact of the COVID-19 lockdown on parents' health-seeking behaviour and care for a sick or injured child in the Netherlands.

**Design and setting** An online survey on parents' experiences with a sick or injured child during the COVID-19 lockdown periods was disseminated through social media.

**Participants** Parents living in the Netherlands with a sick or injured child during the lockdown periods from March to June 2020 and from December 2020 to February 2021 were eligible to participate.

**Outcome measures** Descriptive statistics and thematic analysis were used to analyse family and children's characteristics, parents' response to a sick or injured child, and the perceived impact of the lockdown on child's severity of illness and treatment reported by parents. Analyses were stratified for children with and without chronic conditions.

**Results** Of the 105 parents who completed the survey, 83% reported they would have sought medical help before lockdown compared with 88% who did seek help during lockdown for the same specific medical problem. Parents reported that changes in health services affected their child's severity of illness (31%) and their treatment (39%), especially for children with chronic conditions. These changes included less availability of healthcare services and long waiting lists, which mostly led to worsening of the child's illness. During lockdown, there was no change in health-seeking behaviour by parents of children with a chronic condition (N=51) compared with parents of children without a chronic condition.

**Conclusion** Parents in the Netherlands who completed the survey were not deterred from seeking medical help for their sick or injured child during the COVID-19 lockdown periods. However, changes in health services affected child's severity of illness and treatment, especially for children with chronic conditions.

## BACKGROUND

Globally, healthcare professionals reported that the number of paediatric patients visiting medical services declined significantly while lockdowns were in effect due to the COVID-19

### Strengths and limitations of this study

► Online surveys enable the collection of anonymised data and facilitate the collection of data from a wide range of patients regardless of their residency.
► The survey was disseminated through social media to reach the general population.
► The survey had several 'other' options where parents could give a more detailed explanation of their answers in addition to the multiple-choice answers.
► The advertisement through social media could have caused selection bias as parents who do not have social media or parents with limited (digital) literacy could not have filled in the survey.
► Our study population might not be a good reflection of the general population of ill or injured children as almost 50% of the children had a chronic condition.

pandemic: the estimated decline of paediatric visits to emergency departments (EDs) during lockdown ranged from 30% in the UK to 89% in Italy.[1–4] In the Netherlands, all medical visits to Dutch EDs declined by 25% during the first lockdown from 23 March 2020 to 1 June 2020.[5] The Dutch Healthcare Authority has reported that paediatric urgent referrals by general practitioners (GPs) declined more than referrals by GPs to other specialties.[6] This decline could partially be explained by a lower incidence of illness or injury in children during lockdown as limited contact with others reduced exposure to infectious diseases and restricted activities reduced accidental injuries.[7 8] However, other specialties have shown a recuperation of the number of referrals after the first lockdown, while the number of paediatric referrals has not.[6] The Netherlands had two lockdown periods where civilians were strongly advised to mainly stay at home and schools were closed.

Other explanations for the decline in paediatric medical visits are parental concern about COVID-19 and barriers raised by healthcare professionals.[9] On the one hand, there may be positive outcomes from parents' self-care of their children as they develop ways of coping with and managing their child's illness or injury themselves. On the other hand, it is also important to notice the possible negative outcomes of the decrease in parents seeking help for a sick or injured child. In a survey set out by the Dutch Paediatric Society among 1400 Dutch paediatricians, there were 51 cases reporting unnecessary collateral harm as a result of delayed presentation to medical health services due to the COVID-19 pandemic. These reports include several cases of diabetic ketoacidosis, sepsis and cancer.[10] Similar results were found in surveys set out in other European countries.[9 11 12] Unfortunately, these surveys do not differentiate between delay caused by parents or by healthcare professionals.

It is imperative that we try to unravel the reasoning behind the health-seeking behaviour of parents for ill or injured children and the impact that the COVID-19 lockdown had on their behaviour. Special attention should be given to parents of children with chronic conditions. They may seek less medical care as they are afraid of COVID-19 infection for their vulnerable child. Concerns were also raised regarding the mental health of children and adolescents during these difficult times. Based on a survey in the USA, 35% of adolescents receive mental health support solely in an educational setting.[13] With schools closing, it is possible that certain services could not be accessed properly.[14] Children also ended up spending more time at home and were socially isolated, which may have negatively affected their mental health.[15 16]

We aimed to assess the impact of the COVID-19 lockdown on parents' experiences with a sick or injured child in the Netherlands, stratified for children with and without a chronic condition. In more detail, we focus on whether and where parents sought help and whether and why their health-seeking behaviour had changed due to the lockdown period. In addition, we assessed parents' perceptions of the impact of the lockdown on their child's severity of illness and treatment.

## METHODS
### Study design
This study was a COVID-19-related project in collaboration with the University College London and the University of Plymouth. It was a cross-sectional study consisting of an online survey assessing parents' health-seeking behaviour for, and care of, a sick or injured child during two COVID-19 lockdown periods. Informed consent of the participating parents was obtained.

### Study population and setting
Parents living in the Netherlands who self-identified as having a sick or injured child during the lockdown periods from 23 March 2020 to 1 June 2020 and from

15 December 2020 to 16 February 2021 were included. The main measures taken during these lockdown periods are shown in chronological order in online supplemental appendix A. We excluded parents who did not live in the Netherlands or who did not have a sick or injured child during the aforementioned lockdown periods. Our aim was to recruit about 100 respondents to our survey. For a sample size of 100, the 95% CI for a 50% estimate of proportions is 40.2% to 59.8% to compare the proportion of parents who sought help before lockdown to the proportion of parents that sought help during the lockdown. For a sample size of 50 in each subgroup of children with and without comorbidity the 95% CI for a 50% estimate proportion is 36.2% to 63.8%.[17 18]

### Data collection
The original English questionnaire was translated to the Dutch language. The questions were similar, but the answer options regarding the healthcare system were adapted to the Dutch healthcare system.(online supplemental appendices B and C). The questionnaire was launched as an online survey created in Google Forms, which enabled the collection of anonymised data. Online surveys facilitate the collection of data from a wide range of parents regardless of their residency. The survey was mainly composed of multiple-choice questions, but 'other' options to add free text were available. This 'other' option also enabled parents to give a more detailed explanation on their previous chosen multiple-choice answer. Respondents were recruited using social media and virtual snowball sampling, in which people helped to disseminate the survey to other parents by reposting the survey on their own social media channels. Information about the survey was posted by the research group on Facebook, Twitter, LinkedIn, Instagram, WhatsApp and professional contacts were sent a request to share the information about the study with their social contacts. Facebook was divided into personal Facebook accounts and Facebook groups, of which the latter included private Facebook groups for parents as well as public Facebook pages of the Erasmus MC and the Sophia Children's Hospital. A short introduction to the survey and a link to the survey was written to share on social media (online supplemental appendix D). A reminder was posted on social media every 2 weeks until the end of the survey period. When the number of new respondents stagnated we decided to end the survey period. Key persons who had shared the survey on their social media and their estimated reach were recorded in a document to keep track of the spread of the survey and the number of respondents. There was no incentive provided to parents for participating.

### Patient and public involvement
The public was involved in the design and conduct of the study. Parents have been involved in the project as research team members and reviewed the original survey developed in the UK. In the Netherlands, feedback on the survey and the total duration to fill in the survey came

from two parents who were research team members and three parents not working in the medical field. Additionally, feedback on the survey was given by paediatricians and PhD students of the research group of General Paediatrics Erasmus MC Sophia. However, modifications were not needed and these research group members and parents helped disseminating the survey on social media. We plan to disseminate the results to clinicians and health policy makers so that decisions concerning access to health services and support for parents can be made based on evidence.

Information about the study and its purpose was explained at the beginning of the online survey. The survey introduction also included a statement about the anonymity of the responses, which explained that no personal identifiable data would be collected. At the end of this information section parents were asked to check a statement, which clearly stated that choosing to complete and submit responses to the survey gave consent to their responses being used in our study. Since we were only including parents in the survey and not the children themselves, there were no concerns about capacity to consent. Information on data usage and dissemination was also provided. Lastly, the respondents had the opportunity to contact the research team if they had additional questions or if they wanted to receive a summary of the study findings. Email addresses were provided for both the project lead and an independent paediatrician to whom any concerns or complaints could be directed.

### Data analysis

Data were analysed using SPSS software V.25.0 and free-text data were subjected to thematic analysis. First, descriptive statistics were used for the characteristics of parents and families who completed the survey. Information on living area, access to WiFi, children in the family and whether they have laptops, computers or mobile phones, was collected.

Second, descriptive statistics for children's characteristics, stratified for chronic condition, were used. The following question on chronic conditions was included in the survey: 'Does your child have any pre-existing illnesses, such as chronic or long-term illness, complex needs or a recurring illness?' When parents answered 'yes' they were asked to specify what kind of chronic condition or recurring illness their child had. Children's characteristics, including age, gender, presenting symptoms and hospital admission, were collected as well. Age was categorised into three predefined age groups: <5 years, 5–11 years and 12–17 years. Presenting symptoms were categorised into the following eight subgroups: skin, breathing, body temperature, dehydration, pain, change of behaviour, injury and other.

Third, parents' response to a sick or injured child was described, stratified for chronic condition. The parents were asked what medical symptom(s) their child had during the lockdown and the subsequent questions were (1) whether they would have sought help for the same

problem before lockdown and (2) whether they did seek help for this problem during lockdown. Lastly, we analysed whether changes in health services during the lockdown affected severity of illness of the child and the child's treatment, stratified for chronic condition. McNemar's test and $\chi^2$ tests were used for quantitative analyses with a $p<0.05$ considered as statistically significant. Qualitative analyses using thematic analysis according to Braun and Clarke were used for free text answers from parents on questions regarding the impact of changes in health services on severity of illness of the child and the treatment the child received.[19] Two authors (CT and EL) categorised the free-text answers independently to minimise subjectivity of a single researcher's judgement. A selection of quotes from parents regarding the perceived impact of changes in health services were translated (by CT and EL) and divided into three categories: healthcare services, parents' behaviour and reported healthcare professionals' behaviour. The quotes were categorised into negative and positive experiences perceived by parents.

### RESULTS

A total of 105 respondents filled in the online COVID-19 parent survey in the Netherlands. During the 2-month period from 19 October 2020 to 19 December 2020, 76 respondents completed the survey regarding the first lockdown period. During the second lockdown, an additional 29 respondents completed the survey from 16 January 2021 to 16 February 2021. Figure 1 shows through which social media channel the online survey had reached parents. The online survey reached 60% of the parents via Facebook (personal and groups), 28% via WhatsApp, and 12% via Twitter and LinkedIn.

### Parent and family characteristics

The parent and family characteristics are shown in table 1. Most families came from small cities in an urban area (69%) and nearly half were from the province South-Holland (47%). All parents had access to WiFi, nearly all had laptops or computers (95%), mobile phones (90%) and access to outdoor space during lockdown (97%). The

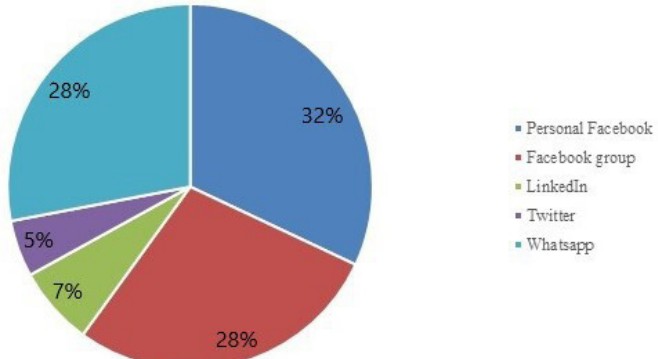

**Figure 1** Pie chart demonstrating through which social media channel the online survey had reached parents.

**Table 1** Parent and family characteristics (N=105)

| | Parent of sick or injured child |
|---|---|
| **Description of living area** | |
| Rural—village | 6 (6) |
| Urban—small city | 72 (69) |
| Urban—big city | 27 (26) |
| **Province** | |
| South Holland | 49 (47) |
| Zeeland | 15 (14) |
| North Brabant | 12 (11) |
| The other seven provinces | 29 (28) |
| Wi-Fi access | 105 (100) |
| Mobile phone | 95 (90) |
| Laptop or computer | 100 (95) |
| Access to outdoor space during lockdown (garden/balcony) | 102 (97) |
| **Where were the school aged children in the family when the illness/injury happened during the lockdown** | |
| All of the children were staying at home | 80 (76) |
| All the children were attending school | 14 (13) |
| Some of the children were attending school | 11 (11) |

Absolute numbers and percentages (%) are shown.

**Table 2** Children's characteristics stratified for chronic condition

| | Total (N=105) | Children with chronic condition (N=51) | Children without chronic condition (N=54) |
|---|---|---|---|
| **Age (years)** | | | |
| <5 | 41 (39) | 12 (24) | 29 (54) |
| 5–11 | 43 (41) | 25 (49) | 18 (33) |
| 12–17 | 21 (20) | 14 (28) | 7 (13) |
| **Gender** | | | |
| Boys | 61 (58) | 28 (55) | 33 (61) |
| Girls | 44 (42) | 23 (45) | 21 (39) |
| **Presenting symptom\*** | | | |
| Skin and appearance | 22 (21) | 11 (22) | 11 (20) |
| Breathing | 25 (24) | 15 (29) | 10 (19) |
| Body temperature | 16 (15) | 9 (18) | 7 (13) |
| Dehydration | 21 (20) | 11 (22) | 10 (19) |
| Pain | 45 (45) | 24 (47) | 23 (43) |
| Change of behaviour | 42 (40) | 21 (41) | 21 (39) |
| Injury | 23 (22) | 8 (16) | 15 (28) |
| Other | 19 (18) | 7 (14) | 13 (24) |
| **Hospital admission** | | | |
| Admitted | 37 (35) | 16 (31) | 21 (39) |

Absolute numbers and percentages (%) are shown.
\*Possible to have more than one presenting symptom.

majority of the school-aged children in the family were staying at home as schools were closed (76%) when the illness or injury happened during lockdown.

## Children's characteristics

In table 2, the children's characteristics are shown for the total population and stratified for chronic condition. Forty-nine per cent of the children had a chronic condition or recurring illness (51/105) such as psychomotor retardation/epilepsy, scoliosis, asthma, eczema or recurring otitis media. The majority of the children were below the age of twelve and were boys. The most common presenting symptoms were pain (45%) and change of behaviour (40%). There was a higher percentage of children aged 12–17 years with chronic conditions compared with children without chronic conditions (28% vs 13%). A higher percentage of children with chronic conditions reportedly had breathing problems (29% vs 19%) and a lower percentage had injuries (16% vs 28%) compared with children without chronic conditions. Other presenting symptoms were, for example, hearing problems, congenital malformations (cleft lip, scoliosis), and facial nerve paralysis. A lower percentage of children with chronic conditions were admitted to the hospital during lockdown compared with children without chronic conditions (31% vs 39%).

## Parents' response to a sick or injured child

Parents reported that 83% (87/105) would have sought help before lockdown, whereas 88% (92/105) of the parents did seek help during lockdown. Health-seeking behaviour of 76 parents during the first lockdown and 29 parents during the second lockdown with comparable restrictions did not differ (data not shown). Four parents reported that they would have sought help before lockdown but did not seek help during lockdown, whereas nine parents would not have sought help before lockdown but did seek help during lockdown. (McNemar's test, p=0.2) (online supplemental appendix E). Thirteen parents (13/105, 12%) who decided not to ask for medical help during lockdown stated that they were not sure if their child was ill or injured enough to need medical help, were worried about themselves or a family member catching COVID-19, thought that the advice 'to stay at home' meant that they could not go to a health centre or hospital or were worried about using health services when they were needed more urgently by other people. Before the lockdown, parents of children with a chronic condition would have sought help for their sick or injured child in 77% (39/51) compared with 89% (48/54) of the

parents of children without a chronic condition ($\chi^2$ test, p=0.1). During the lockdown, parents of children with a chronic condition did seek help for their sick or injured child in 82% (42/51) compared with 93% (50/54) of the parents of children without a chronic condition ($\chi^2$ test, p=0.1). Health-seeking behaviour before and during lockdown related to children's characteristics is shown in online supplemental appendix F. Parents of children below the age of five sought help more frequently and the percentage of parents seeking help decreased with increasing child's age. The percentage of parents seeking help for their sick or injured child was comparable for all presenting symptom groups except for the 'other' presenting symptom group.

The reported medical sources used by parents before lockdown were comparable with the reported sources used during lockdown. The majority of the parents contacted the GP and some parents sought help from their child's paediatrician or another medical professional such as the midwife. A small percentage of the parents called the Dutch emergency number or went to the ED. Parents were asked which other actions they undertook for their sick or injured child. Some parents stated that they chose to wait-and-see whether their child got better. When they did decide to treat the illness or injury themselves, they most often treated their child with paracetamol. Parents sought information on how to treat their child on the internet and by asking their social circle of friends and family. When parents stated that information and advice given were useful, this was mainly because they were reassured that what they were doing was right. A quote from a parent was 'The information was useful since it reassured me, useful tips and information on when to seek medical help (again) were given'. When parents stated that information and advice given were not useful, this was mainly because it was general advice and not child specific. A quote of a parent was 'Every child is unique and requires an individual approach'.

### Impact of changes in health services

Parents reported that changes in health services during lockdown had impact on the severity of illness of their child in 31% (32/105) and on their child's treatment in 39% (41/105). Parents' explanations were mostly negative and included the following responses: long waiting list for diagnostics or treatment, postponed surgery or outpatient clinic consultation, no assessment by a physician, more consultations by telephone, fewer checks and doctor visits during admission. Lastly some parents stated they waited too long to seek help because they were anxious. Contrarily, there were also some positive responses including shorter turnaround time at the ED and more attention from medical staff during admission as the ward was almost empty (online supplemental appendix G). Parents of children with a chronic condition reported that changes in health services affected the severity of their child's illness in 33% (17/51) compared with 28% (15/54) of the parents of children without a

chronic condition ($\chi^2$ test, p=0.5). Parents of children with a chronic condition reported that changes in health services affected their child's treatment in 47% (24/51) compared with 32% (17/54) of the parents of children without a chronic condition ($\chi^2$ test, p=0.1).

## DISCUSSION AND CONCLUSION
### Main findings

Parents in the Netherlands with a sick or injured child during the COVID-19 lockdown and who completed the online survey, were not deterred from seeking medical help for their sick or injured child. Almost 50% of the sick or injured children in our study had a chronic condition or recurring illness. Parents of children without a chronic condition would have sought help more often before lockdown and did seek help more frequently during lockdown than parents of children with a chronic condition. This might be explained by the fact that parents of children with chronic conditions are better able to judge their child's illness or to treat the illness themselves since they have more medical experience. However, more parents of children with a chronic condition reported that changes in health services during lockdown had impact on their child's severity of illness and treatment compared with parents of children without a chronic condition. Parents of children with a chronic condition may have been more anxious during the COVID-19 pandemic about catching COVID-19 themselves or their child, who is perceived to be more vulnerable than a healthy child. The child's age was associated with parents' health-seeking behaviour as parents of children below the age of five sought medical help more frequently, both before and during lockdown. This finding is also described by Sands et al where the majority (70%) of medical attendees in children were below the age of 5.[20] Remarkably, the number of parents who did seek medical help during the lockdown was high in all presenting symptom groups. This might indicate that parents who completed the survey probably had a more severely ill child. This could also explain the high hospital admission rate (35%) in our study. It has been reported that there is an increase in mental health problems in children due to the lockdown.[21] Where breathing problems are a common presenting symptom to seek medical help, 40% of the children in our study had a change of behaviour as presenting symptom which is typical for the lockdown period.[20]

### Strengths and limitations

This online survey was the first national survey in the Netherlands on parents' health-seeking behaviour and care for a sick or injured child during the lockdown period due to the COVID-19 pandemic. The survey was circulated through social media to reach the general population, which was also geographically spread throughout the Netherlands. We could not estimate the reach of the survey since it was reposted by many people on social media through virtual snowball sampling. The

social media channels of our hospital reached more than 10 000 people, but we could not calculate the response rate as we do not know how many of these children were ill or injured in the lockdown periods. The survey had several 'other' options where parents could give a more detailed explanation of their answers in addition to the multiple-choice answers. There were also some limitations. First, the survey was advertised through social media, which could have caused selection bias since parents who do not have access to social media or who are not active on social media could not have filled in the survey. Parents with limited (digital) literacy would also have been unable to fill in the survey. Additionally, this study design was prone to sampling bias. We did not have data on parents with an injured of sick child who did not fill in the survey. It seemed that parents of children with more serious illness had filled in the survey as this was reflected in the high hospital admission rate of 35%. Therefore generalisability might be limited. The majority of the respondents live in the province South-Holland, which could be explained by the fact that Erasmus MC Sophia is located in South-Holland. This study was prone to information bias and recall bias as well. Parents were asked theoretically whether they would have sought help for the same medical problem before lockdown, which was a potential information bias. Recall bias could have occurred when parents where asked about the sickness/injury of their child during the lockdown periods as these events happened before they filled in the questionnaire. However, we assumed that parents who filled in the survey do remember their child's sickness/injury well since these were probably parents of children with more serious illness reflected in the high hospital admission rate. Second, our study population might not be a good reflection of the general population of ill or injured children as almost 50% of the children had a chronic condition or recurring illness. This might be explained by the dissemination strategy or parents of children with a chronic condition being more motivated to participate. However, this allowed us to perform analyses stratified for chronic condition which is independent of our sample of parents who had completed the survey. Furthermore, children with a chronic condition are a vulnerable group, who are more prone to serious illness and this study provides important insights in the health-seeking behaviour of their parents.

### Implications for clinical practice

This study showed that the majority of parents who completed our survey did seek help for a sick or injured child during the lockdown period. However, there were barriers in the health-seeking process such as postponed doctors' visits, no physician's assessment and fewer face-to-face consultations were undertaken as more were performed by telephone. A small group of parents decided not to seek help for their sick or injured child because they were not sure if their child was ill or injured enough to need medical help reflecting prepandemic research findings,[22] were worried about catching COVID-19 or thought that the advice to stay at home meant that they could not attend medical health services. This finding is important since parents should have adequate information on when and where to seek help for a sick or injured child. However, throughout 2020 and 2021, healthcare professionals have been contacting the Dutch media to emphasise the message that EDs and hospitals continue to be available to provide care and treatment for all patients if they need medical help.[23–25]

### CONCLUSION

Parents in the Netherlands who completed the online survey about health-seeking behaviour and care for a sick or injured child during the COVID-19 lockdown were not deterred from seeking medical help for their sick or injured child. However, changes in health services did affect their child's severity of illness and treatment, especially for children with a chronic condition. Although the impact was mostly negative there were some positive consequences as well. These findings are important to ensure that we inform parents, especially of children with chronic conditions, on when and where to seek help during lockdown periods to prevent delayed presentation of children with illness or injury to health services.

**Contributors** CDT and EKL contributed to the conceptualisation and design of the study, coordinated the data collection, carried out the analyses, and drafted the initial and final manuscript. DMB, SN, RC, RBJ, JC and ML contributed to the conceptualisation and design of the study, supervised the analyses and critically reviewed the manuscript. HAM contributed to conceptualisation and design of the study, contributed to interpretation of the data, critically reviewed the manuscript and supervised the study and acts as guarantor. All authors approved the final manuscript as submitted and agree to be accountable for all aspects of the work.

**Funding** This project has received funding from the European Union's Horizon 2020 research and innovation programme under grant agreement No. 848 196.

**Competing interests** None declared.

**Patient consent for publication** Not applicable.

**Ethics approval** The study was approved by the Medical Ethics Committee Erasmus Medical Centre (MEC-2020-0627) and by the Scientific Research Committee SARS-CoV-2 and COVID-19 Erasmus Medical Centre.

**Provenance and peer review** Not commissioned; externally peer reviewed.

**Data availability statement** Data are available on reasonable request. Data from this study are available on request to the corresponding author of the study (c.tan@erasmusmc.nl), subject to local rules and regulations.

**ORCID iDs**
Chantal D Tan http://orcid.org/0000-0002-1148-9716
Ray B Jones http://orcid.org/0000-0002-2963-3421
Dorine M Borensztajn http://orcid.org/0000-0002-2437-0757
Monica Lakhanpaul http://orcid.org/0000-0001-5288-3325
Henriette A Moll http://orcid.org/0000-0001-9304-3322

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
