## [Reviewer comments · BMJ Open]

ARTICLE DETAILS

TITLE (PROVISIONAL)	PARENTS' EXPERIENCES WITH A SICK OR INJURED CHILD DURING THE COVID-19 LOCKDOWN: AN ONLINE SURVEY IN THE NETHERLANDS
AUTHORS	Tan, Chantal; Lutgert, Eveline; Neill, Sarah; Carter, Rachel; Jones, Ray; Chynoweth, Jade; Borensztajn, Dorine; Lakhanpaul, Monica; moll, Henriette

VERSION 1 – REVIEW

REVIEWER	Rees, Chris Boston Children s Hospital, Pediatric Emergency Medicine
REVIEW RETURNED	08-Aug-2021

GENERAL COMMENTS	The authors report the results of a cross-sectional survey conducted online with 105 parents in the Netherlands. They found no reported differences in care seeking during lockdowns in their population. The strengths of this study are the attempt to understand the global phenomenon of reduced clinic/ED visits during lockdown periods from the caregivers' perspective. This is important and, to my knowledge, has not been directly addressed previously. Despite the article's strengths, there are several significant weaknesses that this reviewer thinks should be addressed. First, I worry that the study design and potential for sampling bias may preclude the authors from making some of the claims such as "care seeking did not change during lockdowns". This may be addressed by explaining more about the potential respondents this survey reached. I wonder if the authors can truly say "help seeking behavior had changed during the lockdown period" as this survey was sent during lockdowns and there were no serial surveys sent to truly represent care-seeking behaviors prior to the lockdown periods. I fear that recall bias may preclude the authors from making this claim. At a minimum, this needs to be acknowledged in the Limitations. Abstract: -In the Objective, I believe the authors mean "health seeking behavior" not "help seeking", correct?-I suggest the authors include the response rate (or estimated reach of this survey sent out) in the Abstract to allow readers to understand how representative (or not representative) these findings are of the general population in the Netherlands.-Is the line, "83% reported they would have sought medical help before lockdown compared to 88% who did seek help during lockdown." really a comparison? I think the wording could be
--

	clearer. Would have sought care in general or for a specific problem?  -The phrasing, “changes in health services affected their child’s severity of illness” could be clearer. What type of changes in health services? Less availability? And which direction was their child’s illness severity affected? Better or worse? Should be explicit. -Was “more consultations by telephone” really a negative thing overall? For minor issues, caregivers may be happy to avoid a visit to a provider if a telephone call could replace such an encounter. Background:  -The first sentence needs references given its declarative nature. -Is there any data the authors can point to in order to describe any declines in the incidence of injury related ED visits possibly due to restrictions on sports and other physical activities? Methods:  -I’m not sure it is correct to call this an “observational study” as this is simply collecting data on opinions. I suggest changing this to cross-sectional study. -Page 6, line 39: how did the authors measure that caregivers had a “sick or injured child”? This is very important as this is seemingly an inclusion criteria. It seems it was one of the first questions in the survey. This just needs to be clarified that it was “self identification of having a sick or injured child.” -Page 6, line 49: I appreciate that the authors included their power calculation, but what was their a priori outcome that they aimed to achieve a 50% estimate for? It seems the study is under-powered to conduct any sub-group analyses. -Page 7, line 12: what is snowballing? This should be defined. -Page 7, line: every two weeks until when? -Was any software used to conduct the thematic analysis? -It appears the included surveys are still specific to the UK. Can the authors include the actual survey sent to participants in the Netherlands? -Was any incentive provided to parents for participation? Results:  -Page 10, line 28: was any statistical test done to support the claim that “children with chronic conditions were older and were more often reported to have breathing problems...”? The same comment applies to the sentence beginning on line 33. -Page 11, line 28: I think the authors need to be very explicit that it was a “reported response”. This relates to my comment above about the potential for recall bias in the responses about care seeking before lockdowns. -Page 11, line 47: as above, it seems the study is under-powered to conduct any sub-group analyses, but the authors tried to compare care seeking among caregivers with children with chronic conditions compared to those without chronic conditions. -Tables 3 and 4 could be omitted, in my opinion. All data presented here are in the text. Discussion:  -I think the biggest limitation is the potential for sampling bias as one would presume that a caregiver who did seek care during lockdowns would be more likely to fill out such a survey.
--	---

	-Again, in the limitations paragraph, I don't know that the authors were powered to stratify their analyses. Can the authors explain their power to detect differences between groups?
--	--

REVIEWER	Gibson, Faith Great Ormond Street Hospital For Children NHS Trust, ORCHID
REVIEW RETURNED	24-Aug-2021

GENERAL COMMENTS	Thank you for asking me to review this paper. Reported here is an online survey, using a survey developed and used in the UK. The survey instrument was translated for use, authorship includes UK collaborations. It seeks to examine parents' experiences on health-seeking behaviours and care for sick or injured children during the period of lockdown. I have only a few comments, some style, some content:  1. Background very clear, thank you. 2. There are a couple of places, typographical/translation errors, page 5, line 32, should read 'On the one hand'; Page 6, line 4, should read could not, same page, line 12, missing an a before chronic condition. 3. Page 7, line 10, I wonder if it might read better, recruited using social media and snowballing. 4. Study design, recruitment, data collection etc all very well explained, thank you. 5. Patient and public involvement, I wonder if you can say how many parents looked at the survey, what did you ask them to do, where were this 'group' of parents recruited from? 6. Data analysis is very detailed, but I wonder for ease of reading if there is a way of breaking up this text. You use, first (should be Firstly, to be consistent with your style), secondly etc, to introduce new content, would these be better as new paragraphs. It is clear what you were working with in terms of survey responses, but less clear how much open text you had to work with, bottom of page 8 explains what you did with this free text, but not what you were working with. Same section, line 28, I wonder if some words are missing here.....and hospital admission were also collected, maybe? 7. Staying with the open text, have I missed something, I only see 2 direct quotes, might a table of negative and positive responses, with quotes be a useful addition to your paper. Of course, at the moment we do not know the magnitude of this data, so its not easy to comment, but it feels missing, or maybe just not jumping out, not prioritised in what has been presented: I can see some text page 12-13, is there more that can be said. 8. Page 14, line 49, maybe a word missing, have access to social media. 9. Appendix A is really interesting in terms of context, did any of these factors feature in terms of your analysis/findings as the same or different during the two lockdowns; just wondered? 10. Finally, I wondered about yours plans for how these findings will be shared, using social media, in line with your approach to recruitment. Thank you for considering these suggestions.
---

Comments from reviewer 1:

Comment 1.1: The authors report the results of a cross-sectional survey conducted online with 105 parents in the Netherlands. They found no reported differences in care seeking during lockdowns in their population. The strengths of this study are the attempt to understand the global phenomenon of reduced clinic/ED visits during lockdown periods from the caregivers' perspective. This is important and, to my knowledge, has not been directly addressed previously.

Despite the article's strengths, there are several significant weaknesses that this reviewer thinks should be addressed. First, I worry that the study design and potential for sampling bias may preclude the authors from making some of the claims such as "care seeking did not change during lockdowns". This may be addressed by explaining more about the potential respondents this survey reached. I wonder if the authors can truly say "help seeking behavior had changed during the lockdown period" as this survey was sent during lockdowns and there were no serial surveys sent to truly represent care-seeking behaviors prior to the lockdown periods. I fear that recall bias may preclude the authors from making this claim. At a minimum, this needs to be acknowledged in the Limitations.

Answer 1.1: Thank you for acknowledging the importance of our study. We agree that there are some weaknesses concerning potential biases due to the study design. In our discussion section we mentioned selection bias as potential bias, which we think is in line with the sampling bias you mentioned. The conclusion that care seeking did not change during lockdowns is explicitly concluded based on the parents who have filled in our survey and not in general, since we know that a survey is prone to sampling bias. (See also answer 1.21). Therefore we added the following to the discussion section: *"Additionally, this study design was prone to sampling bias. We did not have data on parents with an injured or sick child who did not fill in the survey. It seemed that parents of children with more serious illness had filled in the survey as this was reflected in the high hospital admission rate of 35%. Therefore generalizability might be limited."* Furthermore, we agree that this study is prone to two other biases, namely information bias and recall bias. We have added the following to the limitations in the discussion section: *"This study was prone for information bias and recall bias as well. Parents were asked theoretically whether they would have sought help before lockdown for the same medical problem, which was a potential information bias. Recall bias could have occurred when parents were asked about the sickness/injury of their child during the lockdown periods as these events happened before they filled in the questionnaire. However, we assumed that parents who filled in the survey do remember their child's sickness/injury well since these were probably parents of children with more serious illness as reflected in the high hospital admission rate."*

- **Abstract**

Comment 1.2: In the Objective, I believe the authors mean "health seeking behavior" not "help seeking", correct?

Answer 1.2: Thank you for your comment. We explicitly use 'help-seeking' as this is the terminology used by parents themselves since they do not refer to health or care seeking. Now it is defined as parents' help seeking behaviour for their sick or injured child, not always from health care services e.g. internet, family.

Comment 1.3: I suggest the authors include the response rate (or estimated reach of this survey sent out) in the Abstract to allow readers to understand how representative (or not representative) these findings are of the general population in the Netherlands.

Answer 1.3: We totally agree that including the response rate would be of additional value. However, it is difficult to estimate the reach of this survey since it was reposted by many people on social media with the virtual snowball sampling. We do have information on the reach of the social media channels of our hospital, namely around 10,000 people, but the difficulty is that we do not know how many of these children were ill or injured in the lockdown periods, which makes it very difficult to estimate the response rate. We added the following in the discussion section: *"We could not estimate the reach of the survey since it was reposted by many people on social media through virtual snowball sampling. The social media channels of our hospital reached more than 10,000 people, but we could not*

calculate the response rate as we do not know how many of these children were ill or injured in the lockdown periods."

Comment 1.4: Is the line, "83% reported they would have sought medical help before lockdown compared to 88% who did seek help during lockdown." really a comparison? I think the wording could be clearer. Would have sought care in general or for a specific problem?

Answer 1.4: Thank you for your comment. These percentages are really a comparison between help seeking behaviour before lockdown and during lockdown for the same specific problem. We have added the following to the methods section: *"The parents were asked what medical symptom(s) their child had during the lockdown and the subsequent questions were 1) whether they would have sought help for the same problem before lockdown and 2) whether they did seek help for this problem during lockdown."* We added 'for the same specific medical problem' in the abstract as following: *"83% reported they would have sought medical help before lockdown compared to 88% who did seek help during lockdown for the same specific medical problem."*

Comment 1.5: The phrasing, "changes in health services affected their child's severity of illness" could be clearer. What type of changes in health services? Less availability? And which direction was their child's illness severity affected? Better or worse? Should be explicit.

Answer 1.5: Thank you for your comment. The questions on this subject were as following: "Have the changes to health services during the lockdown affected how ill your child was and/or how your child was treated?" We took changes in health services as broad as we can, so that in case parents answered 'yes' they could give details in an open text answer on both what and how changes in health services had impact on their child. These open answer questions could have been both positive or negative. We have added the following to the data analysis part of the methods section: *"A selection of quotes from parents regarding the perceived impact of changes in health services were translated (by CT and EL) and divided into three categories: health care services, parents' behaviour and reported health care professionals' behaviour. The quotes were categorized into negative and positive experiences perceived by parents."* See also comment and answer 2.7 (Appendix G).

Comment 1.6: Was "more consultations by telephone" really a negative thing overall? For minor issues, caregivers may be happy to avoid a visit to a provider if a telephone call could replace such an encounter

Answer 1.6: Thank you for your question. We think it could work both ways, but in the majority it was experienced as negative since some parents added that they would have preferred a real life consult. For instance, a parent reported that he/she preferred showing a wound to a doctor in person, rather than sending a photo. See also comment and answer 2.7 (Appendix G).

- **Background**

Comment 1.7: The first sentence needs references given its declarative nature.

Answer 1.7: Thank you for your comment. We have combined the first and second sentence of the background as following as the four references are for this whole first part: *"Globally, health care professionals reported that the number of paediatric patients visiting medical services declined significantly while lockdowns were in effect due to the Coronavirus disease 2019 (COVID-19) pandemic: the estimated decline of paediatric visits to Emergency Departments (EDs) during lockdown ranged from 30% in the United Kingdom to 89% in Italy.(1-4)".*

Comment 1.8: Is there any data the authors can point to in order to describe any declines in the incidence of injury related ED visits possibly due to restrictions on sports and other physical activities?

Answer 1.8: Thank you for your question. We have added reference eight in the background section, which is a study mentioning a decline in injury related ED visits in children due to a stop in practicing sports during the COVID-19 pandemic.

- **Methods**

Comment 1.9: I'm not sure it is correct to call this an "observational study" as this is simply collecting data on opinions. I suggest changing this to cross-sectional study

Answer 1.9: Thank you for your comment. We realize that there is some overlap between these study designs, but we think an observational study better describes our study. A cross-sectional study implies that we collected data at a single point in time in order to compare different population groups.

However, we collected data over 2 different periods of time concerning two separate lockdown periods and we did not compare different population groups.

Comment 1.10: Page 6, line 39: how did the authors measure that caregivers had a “sick or injured child”? This is very important as this is seemingly an inclusion criteria. It seems it was one of the first questions in the survey. This just needs to be clarified that it was “self identification of having a sick or injured child.”

Answer 1.10: Thank you for your comment, we completely agree that this should be clarified. We have adjusted this sentence in the study population and setting methods section: *“Parents living in the Netherlands who self-identified as having a sick or injured child during the lockdown periods from March 23rd until June 1st 2020 and from December 15th 2020 until February 16th 2021 were included.”*

Comment 1.11: Page 6, line 49: I appreciate that the authors included their power calculation, but what was their a priori outcome that they aimed to achieve a 50% estimate for? It seems the study is under-powered to conduct any sub-group analyses.

Answer 1.11: Thank you for your comment. The a priori outcome was a 50% estimate of parents who would have sought help before lockdown to compare it to the proportion of parents who did seek help during the lockdown. We have added the following to the methods section: *“For a sample size of 100, the 95% confidence interval for a 50% estimate of proportions is 40.2% to 59.8% to compare the proportion of parents who sought help before lockdown to the proportion of parents that sought help during the lockdown.”* For the subgroup analyses we have added the following sentence: *“For a sample size of 50 in each subgroup of children with and without comorbidity the 95% confidence interval for a 50% estimate proportion is 36.2% to 63.8%.”*

Comment 1.12: Page 7, line 12: what is snowballing? This should be defined.

Answer 1.12: Thank you for your comment. We have changed the wording into virtual snowball sampling and adjusted this sentence as following: *“Respondents were recruited using social media and virtual snowball sampling, in which people helped disseminating the survey to other parents by reposting the survey on their own social media channels.”*

Comment 1.13: Page 7, line: every two weeks until when?

Answer 1.13: We have adapted this sentence as following: *“A reminder was posted on social media every two weeks until the end of the survey period. When the number of new respondents stagnated we decided to end the survey period.”*

Comment 1.14: Was any software used to conduct the thematic analysis?

Answer 1.14: Thank you for your question. We did not use any specific software to conduct the thematic analysis since we used thematic analysis for the open text answers on two questions. In the case where parents answered that changes in health services affected the severity of illness and/or the treatment of their child during lockdown we used thematic analysis.

Comment 1.15: It appears the included surveys are still specific to the UK. Can the authors include the actual survey sent to participants in the Netherlands?

Answer 1.15: Thank you for your comment. The actual survey sent to participants in the Netherlands can be found in Appendix C. It was a translated version of the original UK survey with some adaptations made regarding available health care services in the Netherlands.

Comment 1.16: Was any incentive provided to parents for participation?

Answer 1.16: We added the following to the methods section: *“There was no incentive provided to parents for participating.”*

- **Results**

Comment 1.17: Page 10, line 28: was any statistical test done to support the claim that “children with chronic conditions were older and were more often reported to have breathing problems...”? The same comment applies to the sentence beginning on line 33.

Answer 1.17: No statistical tests were performed for table 2, since it is a descriptive table of the children's characteristics. We have adjusted the text as following: *"There was a higher percentage of children aged 12-17 years with chronic conditions compared to children without chronic conditions (28% versus 13%). A higher percentage of children with chronic conditions reportedly had breathing problems (29% versus 19%) and a lower percentage had injuries (16% versus 28%) compared to children without chronic conditions. A lower percentage of children with chronic conditions were admitted to the hospital during lockdown compared to children without chronic conditions (31% versus 39%)."*

Comment 1.18: Page 11, line 28: I think the authors need to be very explicit that it was a "reported response". This relates to my comment above about the potential for recall bias in the responses about care seeking before lockdowns.

Answer 1.18: We agree that we should emphasize that it was a reported response. However, we deleted this sentence since we have omitted table 3.

Comment 1.19: Page 11, line 47: as above, it seems the study is under-powered to conduct any subgroup analyses, but the authors tried to compare care seeking among caregivers with children with chronic conditions compared to those without chronic conditions.

Answer 1.19: We agree that the number of children with chronic conditions are small, but for a sample size of 50 in each subgroup the 95% confidence interval for a 50% estimate proportion is 36.2% to 63.8%. See also comment and answer 1.11.

Comment 1.20: Tables 3 and 4 could be omitted, in my opinion. All data presented here are in the text

Answer 1.20: Thank you for your suggestion. We agree that the data presented in tables 3 and 4 are described in the text as well. Therefore we have omitted these two tables.

- **Discussion**

Comment 1.21: I think the biggest limitation is the potential for sampling bias as one would presume that a caregiver who did seek care during lockdowns would be more likely to fill out such a survey.

Answer 1.21: Thank you for your comment. We agree that sampling bias might influence our results. We have added this limitation to the discussion section as follows: *"This study design was prone to sampling bias. We did not have data on parents with an injured or sick child who did not fill in the survey. Additionally, it seemed that parents of children with more serious illness have filled in the survey as this was reflected in the high hospital admission rate of 35%. Therefore generalizability might be limited."* See also comment and answer 1.1.

Comment 1.22: Again, in the limitations paragraph, I don't know that the authors were powered to stratify their analyses. Can the authors explain their power to detect differences between groups?

Answer 1.22: Thank you for your comment. We described details in previous comment and answer 1.11 and 1.19.

Comments from reviewer 2:

Thank you for asking me to review this paper. Reported here is an online survey, using a survey developed and used in the UK. The survey instrument was translated for use, authorship includes UK collaborations. It seeks to examine parents' experiences on health-seeking behaviours and care for sick or injured children during the period of lockdown. I have only a few comments, some style, some content:

Comment 2.1: Background very clear, thank you.

Answer 2.1: Great to hear that the background was very clear.

Comment 2.2: There are a couple of places, typographical/translation errors, page 5, line 32, should read 'On the one hand'; Page 6, line 4, should read could not, same page, line 12, missing an a before chronic condition.

Answer 2.2: Thank you for noticing some typographical/translation errors in the background section. We have adjusted the following: *"On the one hand", "could not", "a chronic condition"*.

Comment 2.3: Page 7, line 10, I wonder if it might read better, recruited using social media and snowballing.

Answer 2.3: Thank you for your suggestion, we totally agree. We adjusted this sentence as following in the methods section: *“Respondents were recruited using social media and virtual snowball sampling, in which people helped to disseminate the survey to other parents by reposting the survey on their own social media channels”*.

Comment 2.4: Study design, recruitment, data collection etc all very well explained, thank you.

Answer 2.4: Thank you for the compliments.

Comment 2.5: Patient and public involvement, I wonder if you can say how many parents looked at the survey, what did you ask them to do, where were this ‘group’ of parents recruited from?

Answer 2.5: We have added information on the parents who were involved in the design and conduct of the study. The following text was added to the patient and public involvement section: *“Parents have been involved in the project as research team members and reviewed the original survey developed in the United Kingdom. In the Netherlands, feedback on the survey and the total duration to fill in the survey came from two parents who were research team members and three parents not working in the medical field.”*

Comment 2.6: Data analysis is very detailed, but I wonder for ease of reading if there is a way of breaking up this text. You use, first (should be Firstly, to be consistent with your style), secondly etc, to introduce new content, would these be better as new paragraphs. It is clear what you were working with in terms of survey responses, but less clear how much open text you had to work with, bottom of page 8 explains what you did with this free text, but not what you were working with. Same section, line 28, I wonder if some words are missing here.....and hospital admission were also collected, maybe?

Answer 2.6: We agree that the data analysis section is very detailed and that we should break up the text for ease of reading. We have changed ‘first’ into *“firstly”* and started new paragraphs for the sentences starting with secondly and thirdly. The survey had several open text answer options allowing parents to give additional explanation depending on what parents answered the question before as shown in appendix B. The free text answers were used for information concerning the impact of changes in health services on severity of illness of the child and the treatment the child received. See also comment and answer 1.5. The most common free text answers on these questions are summarized as translated quotes in appendix G (shown in comment and answer 2.7). We rephrased the sentence about children’s characteristics in the data analysis section as following: *“Children’s characteristics, including age, gender, presenting symptoms and hospital admission, were collected as well”*.

Comment 2.7: Staying with the open text, have I missed something, I only see 2 direct quotes, might a table of negative and positive responses, with quotes be a useful addition to your paper. Of course, at the moment we do not know the magnitude of this data, so its not easy to comment, but it feels missing, or maybe just not jumping out, not prioritised in what has been presented: I can see some text page 12-13, is there more that can be said.

Answer 2.7: Thank you for your suggestion. We agree that including more open text responses would be interesting. We have added two tables in appendix G with the most given positive and negative responses of parents on the questions concerning how changes in health services influenced the severity of illness of their child and the treatment their child received. We made three main categories including health care services, parents’ behaviour and reported health care professionals’ behaviour.

We have added Appendix G in the section concerning impact of changes in health services in the results section. See also comment and answer 1.5.

Appendix G – Selection of quotes from parents

Severity of illness – quotes based on 32 open text answers

Positive experiences	Negative experiences
Health care services	

“Normally the waiting time at the emergency department can take hours. However, now we were seen and treated within 30 minutes. This spared my child hours of pain !!! Thus a very positive experience!!!!”	“My son had to undergo surgery in a tertiary hospital. Normal waiting time would be 2 weeks, but now it was extended to 1.5 month. During these times, he cried a lot and was in a lot of pain; pure hell.”
“She underwent surgery on her heart defect during the lockdown.”	“The doctor couldn’t see my son, so the severity of the wound wasn’t clear. The advice “to call back in a few days” wasn’t of much help...”
	“Because my child was not seen at the GP out of office for his double ear infection, he received his antibiotic treatment later.”
Parents’ behaviour	
	“We waited a day / night too long to seek medical help out of fear.”
Reported health care professionals’ behaviour	
“The emergency department was empty and we were nearly the only ones at the paediatric ward. The doctors had a lot of time for us.”	“They wanted the patients to leave as soon as possible.”

Comparable quotes are not separately described

Treatment – quotes based on 39 open text answers

Positive experiences	Negative experiences
Health care services	
“ All elective surgeries were postponed and the paediatric ward has never been this empty. We received the best medical care during this period. There was enough time to properly start palliative care.”	“My son needed to wait much longer on his surgery than if there had not have been a lockdown.”
	“My child did not undergo surgery but received a higher dose of antibiotics.”
Parents’ behaviour	
	“We waited longer to consult the GP. Normally we would have contacted the GP earlier, but we did not out of fear of getting infected with COVID-19 and the impact of the protective clothing health care workers wore on our child.”
Reported health care professionals’ behaviour	
	“ We have spent hours in the emergency department in an isolated room without any communication. The nurses gave us little attention and we were asked to take care of the

	nightshift ourselves because it was difficult for the nurses to change into protective clothing. Reason for us to go home the next day.”
	“The doctor could not see us for a consultation. I felt unheard and not understood.”
	“Our GP was very worried, leading to daily consultations while it all started with a simple fever.”

Comparable quotes are not separately described

Comment 2.8: Page 14, line 49, maybe a word missing, have access to social media.

Answer 2.8: Thank you for noticing. We have added the words ‘access to’ in the discussion section: “*who do not have access to social media*”.

Comment 2.9: Appendix A is really interesting in terms of context, did any of these factors feature in terms of your analysis/findings as the same or different during the two lockdowns; just wondered?

Answer 2.9: Thank you for your interesting suggestion. We agree that it is interesting to compare the two lockdown periods, despite the large difference in sample size during the two lockdown periods. The main restrictions during these two lockdown were comparable with social distancing and closing of schools and restaurants. When comparing help seeking behaviour of the 76 parents during the first lockdown and the 29 parents during the second lockdown, we found the following results:

- First lockdown: 86% (65/76) would have sought help before lockdown, 89% (68/76) did seek help during lockdown

- Second lockdown: 76% (22/29) would have sought help before lockdown, 83% (23/29) did seek help during lockdown.

Thus in both lockdowns parents were not deterred from seeking medical help for their sick or injured child. We have added the following to the section parents’ response to a sick or injured child in the results section: “*Help seeking behaviour of 76 parents during the first lockdown and 29 parents during the second lockdown with comparable restrictions did not differ (data not shown)*”.

Comment 2.10: Finally, I wondered about yours plans for how these findings will be shared, using social media, in line with your approach to recruitment.

Answer 2.10: Thank you for your comment. At the time writing, the Netherlands was in the midst of a third lockdown (June 2021). This highlights the importance of these findings being shared with health care professionals to be aware of parents’ help seeking behaviour for their sick or injured child during lockdown periods. We will share this paper after publication on social media channels of Erasmus MC and Sophia Children’s Hospital to hopefully reach a lot of parents and the parents who filled in the survey.

VERSION 2 – REVIEW

REVIEWER	Rees, Chris Boston Children s Hospital, Pediatric Emergency Medicine
REVIEW RETURNED	31-Oct-2021
GENERAL COMMENTS	Thank you for the opportunity to review this revised manuscript. The authors have been very responsive to my initial comments. I appreciate their attentive responses. I think the limitations are now clearly spelled out and acknowledged. My comments are minor at this point and, seemingly, were not addressed in the initial revision. Abstract:

	-In the Objective, I believe the authors mean “health seeking behavior” not “help seeking”, correct? -The phrasing, “changes in health services affected their child’s severity of illness” could be clearer. What type of changes in health services? Less availability? And which direction was their child’s illness severity affected? Better or worse? Should be explicit. -Was “more consultations by telephone” really a negative thing overall? For minor issues, caregivers may be happy to avoid a visit to a provider if a telephone call could replace such an encounter. Methods: -I’m not sure it is correct to call this an “observational study” as this is simply collecting data on opinions. I suggest changing this to cross-sectional study. -It appears the included surveys are still specific to the UK. Can the authors include the actual survey sent to participants in the Netherlands? Thank you again for allowing me to review this interesting work. I hope my comments help the authors convey their message in a rigorous and helpful way.
--	---

REVIEWER	Gibson, Faith Great Ormond Street Hospital For Children NHS Trust, ORCHID
REVIEW RETURNED	31-Oct-2021

GENERAL COMMENTS	Thank you for attending to all reviewers comments, offering up responses, adding to discussion points. I feel the authors have attended to all concerns, certainly my comments have all been addressed, thank you.
--

VERSION 2 – AUTHOR RESPONSE

Reviewer: 1

Dr. Chris Rees, Boston Children’s Hospital

Comment 1.1:

Thank you for the opportunity to review this revised manuscript. The authors have been very responsive to my initial comments. I appreciate their attentive responses. I think the limitations are now clearly spelled out and acknowledged. My comments are minor at this point and, seemingly, were not addressed in the initial revision. Thank you again for allowing me to review this interesting work. I hope my comments help the authors convey their message in a rigorous and helpful way.

Answer 1.1: Dear Dr. Chris Rees, thank you for reviewing our manuscript. Great to hear that the limitations are now clearly spelled out. Since the comments for this minor revision were not addressed detailed enough in our first rebuttal letter, we have added details in our responses below. We hope we have addressed your comments sufficiently in this rebuttal letter.

Abstract:

Comment 1.2: In the Objective, I believe the authors mean “health seeking behavior” not “help seeking”, correct?

Answer 1.2: Thank you for your comment. We used ‘help seeking behaviour’ as this is the terminology used by parents themselves since they do not refer to health or care seeking. However, in literature the terminology of health seeking behaviour is commonly used. Therefore, we have changed help seeking behaviour into health seeking behaviour throughout the abstract and manuscript as you suggested.

Comment 1.3: The phrasing, “changes in health services affected their child’s severity of illness” could be clearer. What type of changes in health services? Less availability? And which direction was their child’s illness severity affected? Better or worse? Should be explicit.

Answer 1.3: Thank you for your comment. Parents mentioned changes in health services such as less availability of health care services and long waiting lists. We have added the following to the abstract: “*These changes included less availability of health care services and long waiting lists, which mostly led to worsening of the child’s illness.*” The child’s severity of illness was mostly affected negatively, but parents did also had some positive experiences which led to better outcome as described in appendix G. This appendix G shows a selection of quotes from parents stratified for both positive and negative experiences.

Comment 1.4: Was “more consultations by telephone” really a negative thing overall? For minor issues, caregivers may be happy to avoid a visit to a provider if a telephone call could replace such an encounter

Answer 1.4: Thank you for your question. We think it could work both ways, but in our study it was experienced as negative since parents added that they would have preferred a real life consult. For instance, a parent reported that he/she preferred showing a wound to a doctor in person, rather than sending a photo. See also Appendix G. However, we have deleted the following sentence in the abstract: “Negative experiences included long waiting lists, delayed referral and more consultations by telephone.” We replaced it with the sentence mentioned above in answer 1.3: “*These changes included less availability of health care services and long waiting lists, which mostly led to worsening of the child’s illness.*”

Methods:

Comment 1.5: I’m not sure it is correct to call this an “observational study” as this is simply collecting data on opinions. I suggest changing this to cross-sectional study

Answer 1.5: Thank you for your comment. We agree that there is some overlap between these study designs. Therefore, we have adjusted this in the methods section: “*It was an cross-sectional study...*”

Comment 1.6: It appears the included surveys are still specific to the UK. Can the authors include the actual survey sent to participants in the Netherlands?

Answer 1.6: Thank you for your comment. The actual survey sent to participants in the Netherlands can be found in Supplementary File 3 - Appendix C. It was a translated version of the original UK survey. The main changes made were regarding the health care system. The questions were similar, but we made changes in the answer options regarding the health care system and in which area people lived in. For instance, the answer options in the UK survey on where parents sought medical help included NHS Direct 111, GP surgery and minor injuries unit. These services do not exist in the Netherlands and were therefore deleted in the Dutch survey. Another example is the question regarding where parents looked for information on how to manage the illness or injury. Answer options in the UK survey that were not applicable for the Dutch health care system and were therefore deleted, included NHS app and NHS choices. For the Dutch survey we added the answer option thuisarts.nl, which is a commonly used website for patients with information on how to manage specific symptoms. We have added the following sentence to the data collection part in the methods section: “*The questions were similar, but the answer options regarding the health care system were adapted to the Dutch health care system.*”

Comment 1.7: Thank you again for allowing me to review this interesting work. I hope my comments help the authors convey their message in a rigorous and helpful way.

Answer 1.7: Thank you for your comments and suggestions, it has improved our paper significantly. We hope we have addressed your comments sufficiently and that we gave enough details in our responses.

Reviewer: 2

Prof. Faith Gibson, Great Ormond Street Hospital For Children NHS Trust, University of Surrey

Comment: Thank you for attending to all reviewers comments, offering up responses, adding to discussion points. I feel the authors have attended to all concerns, certainly my comments have all been addressed, thank you.

Answer: Dear Prof. Faith Gibson, thank you for your comments. We think addressing these comment has improved our paper significantly.